# PCR Array Profiling of miRNA Expression Involved in the Differentiation of Amniotic Fluid Stem Cells toward Endothelial and Smooth Muscle Progenitor Cells

**DOI:** 10.3390/ijms25010302

**Published:** 2023-12-25

**Authors:** Florin Iordache, Adriana (Ionescu) Petcu, Aurelia Magdalena Pisoschi, Loredana Stanca, Ovidiu Ionut Geicu, Liviu Bilteanu, Carmen Curuțiu, Bogdan Amuzescu, Andreea Iren Serban

**Affiliations:** 1Department of Preclinical Sciences, Faculty of Veterinary Medicine, University of Agronomic Sciences and Veterinary Medicine of Bucharest, 105 Blvd. Splaiul Independentei, 050097 Bucharest, Romania; ionescu.adriana95@gmail.com (A.P.); aureliamagdalenapisoschi@yahoo.ro (A.M.P.); loredana.stanca@fmvb.usamv.ro (L.S.); geicu.ovidiu@gmail.com (O.I.G.); liviu.bilteanu@gmail.com (L.B.); irensro@yahoo.com (A.I.S.); 2S.C. Personal Genetics S.R.L. Genetic Medical Center, 010987 Bucharest, Romania; 3Department of Microbiology and Immunology, Faculty of Biology, University of Bucharest, 91–95 Splaiul Independentei, 050095 Bucharest, Romania; carmen.curutiu@bio.unibuc.ro; 4Department of Biophysics and Physiology, Faculty of Biology, University of Bucharest, 91–95 Splaiul Independentei, 050095 Bucharest, Romania; bogdan@biologie.kappa.ro

**Keywords:** microRNA, stem cells, differentiation, endothelial progenitor cells, smooth muscle progenitor cells

## Abstract

Differentiation of amniotic fluid stem cells (AFSCs) into multiple lineages is controlled by epigenetic modifications, which include DNA methylation, modifications of histones, and the activity of small noncoding RNAs. The present study investigates the role of miRNAs in the differentiation of AFSCs and addresses how their unique signatures contribute to lineage-specific differentiation. The miRNA profile was assessed in AFSCs after 4 weeks of endothelial and muscular differentiation. Our results showed decreased expression of five miRNAs (miR-18a-5p, miR-125b-5p, miR-137, miR-21-5p, and let-7a) and increased expression of twelve miRNAs (miR-134-5p, miR-103a-3p, let-7i-5p, miR-214-3p, let-7c-5p, miR-129-5p, miR-210-3p, let-7d-5p, miR-375, miR-181-5p, miR-125a-5p, and hsa-let-7e-5p) in endothelial progenitor cells (EPCs) compared with undifferentiated AFSCs. AFSC differentiation into smooth muscle revealed notable changes in nine out of the 84 tested miRNAs. Among these, three miRNAs (miR-18a-5p, miR-137, and sa-miR-21-5p) were downregulated, while six miRNAs (miR-155-5p, miR-20a-5p, let-7i-5p, hsa-miR-134-5p, hsa-miR-214-3p, and hsa-miR-375) exhibited upregulation. Insights from miRNA networks promise future advancements in understanding and manipulating endothelial and muscle cell dynamics. This knowledge has the potential to drive innovation in areas like homeostasis, growth, differentiation, and vascular function, leading to breakthroughs in biomedical applications and therapies.

## 1. Introduction

Amniotic fluid cells are ideal candidates for regenerative medicine applications not only due to their high differentiation capacity to various lineages, but also because of their immunomodulatory and wound healing effects, absence of tumorigenic risk, and lack of ethical and legal limitations associated with other stem cell preparations [1]. Notably, amniotic fluid contains a heterogeneous population of mesenchymal cells with fetal origins, exclusively presenting HLA class I antigens on their surfaces, hence reducing the likelihood of triggering an immune response [2]. Cultured amniotic fluid cells can adopt different phenotypes including those resembling epithelial and fibroblast cells. About 0.1–0.5% of stem cells are defined by c-kit (CD117+) expression. The renewing potential of amniotic fluid-derived mesenchymal stem cells (AFSCs) emerges from their capacity to differentiate and reduce immune and inflammatory responses at the implantation site [3]. Amniotic fluid stem cells are able to differentiate into various cell lines such as cardiac, endothelial, neural, and osteoblastic lines [4,5,6,7]. Specific differentiation protocols and conditioning media have been devised to culture AFSCs, but seemingly non-mesenchymal derivatives could be obtained only from c-kit positive stem cell clones isolated from amniotic fluid cell pools [4]. When grown in hydrogels, AFSCs have the potential to form blood vessel-like networks [7]. Furthermore, when AFSCs are indirectly cultivated with cardiac cells, cardiac regeneration is stimulated due to paracrine signaling [5,6,7]. Epigenetic alterations have a substantial role in deciding the fate of AFSCs in terms of differentiation into various lineages. The epigenetic control of stem cells differentiation is leveraged via several processes such as DNA methylation, histone alterations, and the active involvement of small noncoding RNAs; however, the precise mechanisms are, so far, still unclear [8]. MicroRNAs (miRNAs) are a subset of small noncoding RNAs, usually 19 to 30 base pairs long in their mature, single-stranded configuration, playing a key role in modulating diverse biological processes through the regulation of post-transcriptional modifications in specific genes [9]. MiRNA can control hundreds of messenger RNAs involved in both adaptive and innate immune responses, as well as in growth, differentiation, apoptosis, and cell senescence [10]. The initial investigations of the role of miRNAs in AFSC differentiation indicated a notable capacity of AFSCs to differentiate into chondrogenic and osteogenic cellular lines. However, adipocyte differentiation did not occur, as evidenced by the expression of miR-21, primarily associated with osteogenic differentiation [11,12]. Subsequent studies showed that miR-let7g and miR-302a might be involved in interaction with POU5F1 and Nanog in processes like differentiation, osteogenesis, and adipogenesis [13]. Glemžaitė and Navakauskienė highlighted that osteogenic differentiation of AFSCs is mediated by specific microRNA expression (miR-17 and miR-148b), chromatin-modifying enzymes, and histone modifications (H3K9ac, H4 hyperAc, and H3K27me3) [14]. Current literature data shows that miRNAs can control the MAPK, WNT, and TGF-β signaling pathways, thus playing a crucial role in the fate of AFSCs [15]. Analysis of cultured AFSCs between passages 1 and 15 by Miranda-Sayago et al., proved major variation regarding the expression of tested miRNAs, generally related to a decrease in expression of TP53 and an increase in expression of miR-125a, which is an indicator of proliferation and stemness [16]. Zentelytė et al. showed that human mesenchymal stem cells isolated from amniotic fluid express miR-146a, miR-21, miR-17, and miR-34a. Furthermore, histone H3K4/H3K9 was trimethylated and the levels of DNMT1, HDAC1, and PRC1/2 proteins decreased. They observed that no significant correlation could be identified between AFSCs isolated from normal gestation vs. pregnancy in which the fetus presents different anomalies and between those differentiated toward adipocytes or osteoblasts [17]. The same group also demonstrated that neural and myogenic differentiation corelates with the down-regulation of miR-17 and miR-21 and up-regulation of miR-146a, miR-34a, and DNMT3, as well as a reduction of the protein expression of DNMT1, HDAC1, H3K9me3, and PRC1/2 (BMI1/SUZ12, EZH2) [18]. Notwithstanding, a more profound comprehension of the underpinnings of AFSCs multi-lineage potential is essential for the effective utilization of these cells in therapeutic applications [19,20]. The present study investigates the role of miRNAs in AFSC differentiation and addresses how their unique signatures contribute to lineage-specific differentiation.

## 2. Results

### 2.1. AFSCs Differentiated to Endothelial Progenitor Cells (EPCs) and Smooth Muscle Progenitor Cells (SPCs)

After 4 weeks of stimulation with 10 ng/mL basic fibroblast growth factor (bFGF), 40 ng/mL vascular endothelial growth factor (VEGF), 10 ng/mL epidermal growth factor (EGF), 20 ng/mL insulin growth factor (IGF-1), in M200 medium, cultures of adherent AFSCs started to form colonies with epithelial-like “cobblestone” morphology (Figure 1A). The differentiation of AFSCs was validated by representative functional characteristics of endothelial progenitor cells. Cultured on Matrigel for 24 h, EPCs formed vascular-like structures (Figure 1B), which bound ac-LDL and *Ulex europaeus* lectin tracers upon incubation (Figure 1G–I). The gene expression assay showed increased mRNA expression for: CD31 (151-fold), eNOS (106-fold), CD144 (26-fold), and wVF (6.6-fold) (Figure 1C). Confirmation of differentiation of AFSCs to endothelial progenitor cells was revealed by immunofluorescence staining for the detection of endothelial cell-specific markers such as CD31, VE-cadherin, and VEGFR2 (Figure 1D–F). The endothelial differentiation was also demonstrated by a characteristic immunophenotypic profile of EPCs that was demonstrated in our previous work [21].

Stimulation of AFSCs with EGF (epidermal growth factor), bFGF (basic fibroblast growth factor), heparin, IGF (insulin growth factor), and BSA (bovine serum albumin) promoted differentiation toward smooth muscle cells, inducing some phenotypic changes such as spindle shape morphology and elongated actin-rich protrusions, which are shown in Figure 2A. The smooth muscle differentiation of AFSCs was confirmed by the upregulation of mRNA expression of smoothelin (50-fold), troponin (13-fold), calponin-1 (4.7-fold), smooth muscle actin (2.7-fold), caldesmon (2.08-fold), cav3.1 (2-fold), and myosin heavy chain 11 (1.47-fold) (Figure 2B). Whole-cell patch-clamp experiments in the classical (ruptured) or β-escin-perforated configuration were performed on a total number of 24 SPCs, with application of different pharmacological compounds (Appendix A). All cells featured outward-rectifying K^+^ currents. BK currents (with standard deviation of current fluctuations measured over the last 250 ms of the depolarizing step at +60 mV exceeding 5 pA) were present in 91.7% of the cells, while inactivatable outward K^+^ currents were present in 41.7% of cells), with *I*_Na_ only being recorded in one cell, a major difference compared to previous results on cultured AFSCs [21,22]. The inactivatable outward K^+^ current component was partly inhibited by phrixotoxin-1 (Appendix A), proving it was produced by Kv4.2/4.3 channels, while other inhibitors (DIDS, glibenclamide, NaCN) did not exert significant effects. The patch-clamp experiments demonstrated the presence in SPCs of calcium-activated potassium channels with big conductance (BK channels—K_Ca_1.1), producing an outward current component at positive potential when applying a double-ramp protocol (Figure 2C), which could be blocked by iberiotoxin 100 nM. Furthermore, in 1 of 24 cells, we recorded two inward current components with different activation thresholds and I–V plots suggestive of *I*_CaT_ and *I_CaL_*, and in another cell, only the *I*_CaT_ component. The fast component was resistant to application of nifedipine 1 μM, but it was inhibited by subsequent application of mibefradil 10 μM (Appendix A). These results suggest the differentiation of AFSCs into SPCs (Figure 2D,E). The presence of T-type calcium channels was confirmed by the mRNA expression levels of cav3.1.

### 2.2. MiRNA Regulates the Differentiation of AFSCs

The investigation of the miRNA profiles using the Human Cell Differentiation and Development miRNA Array in AFSCs after 4 weeks of endothelial and muscular differentiation stimulation revealed changes in miRNA expression. Among the 84 arrayed miRNAs, EPCs derived from AFSCs showed increased expression of let-7i-5p (17.39-fold), miR-375 (11-fold), miR-134-5p (7.62-fold), miR-129-5p (7.46-fold), let-7c-5p (7.36-fold), miR-210-3p (7.11-fold), miR-103a-3p (4.89-fold), let-7d-5p (4.86-fold), miR-214-3p (4.20-fold), miR-181a-5p (2.75-fold), and miR-125a-5p (2.75-fold). Decreased expression was revealed for miR-125b-5p (−8.57-fold), let-7e-5p (−3.48-fold), let-7a-5p (−2.83-fold), miR-18a-5p (−2.69-fold), miR-137 (−2.07-fold), and miR-21-5p (−2.03-fold) (Figure 3).

SPCs derived from AFSCs showed increased expression of miR-155-5p (4.38-fold), let-7i-5p (3.78-fold), miR-20a-5p (3.16-fold), miR-214-3p (2.48-fold), miR-134-5p (2.33-fold), and miR-375 (2.17-fold) and downregulation of miR-18a-5p (−12.82-fold), miR-137 (−9.85-fold), and miR-21-5p (−2.28-fold) (Figure 4).

## 3. Discussion

Recent miRNA studies established a signature that is typical for embryonic stem cells [19,20]. The miRNA clusters act as cell cycle mediators increasing embryonic stem cell proliferation, promoting transition from the G phase to S1 phase, while other families, such as miR-291-3p, miR-295 and miR-294, control pluripotency factors (Oct4, Sox2, and Nanog). By acting on these transcription factors they promote somatic reprogramming, activating mesenchymal-to-epithelial changes necessary for colony formation, differentiation, and inhibition of cellular senescence [21]. Our results showed decreased expression of hsa-miR-18a-5p, miR-125b-5p, hsa-miR-137, hsa-miR-21-5p, and hsa-let-7a-5p in EPCs compared with undifferentiated AFSCs (Figure 5). These results agree with other research showing that a low expression level of miR-125 in endothelial cells may be the “youth code” that maintains the normal function of human endothelial cells. The increased expression of miR-125 in endothelial cells of senescent mice can regulate angiogenesis by targeting RTEF-1 and stimulating the expression of eNOS and VEGF [22]. Furthermore, let-7e controls tube formation and migration of endothelial progenitor cells via targeting FASLG. Let-7e overexpression activates Th1 and Th17 cells and regulates the inflammatory responses of human endothelial cells by targeting different inflammatory genes [23]. Among the 84 arrayed miRNAs, EPCs derived from AFSCs showed increased expression in 12 miRNAs (miR-134-5p, let-7i-5p, let-7d-5p, ha-let-7e-5p, miR-103a-3p, miR-214-3p, let-7c-5p, miR-129-5p, miR-210-3p, miR-125a-5p, miR-375, and miR-181-5p) (Figure 5). Zang et al. showed that vildagliptin (dipeptidyl peptidase 4 inhibitor) attenuated endothelial dysfunction in diabetic rats by inhibiting the expression of miR-134-5p that stimulates BDNF and regulates endothelial function [24]. MiR-103a-3p promotes EPC migration and angiogenesis, being downregulated in EPCs isolated from patients with deep vein thrombosis. Downregulation of miR-103a-3p is involved in endothelial progenitor cell dysfunction by activation of PTEN protein [25]. Yamakuchi’s profiling of microRNA in aortic endothelial cells showed that the miR-21, miR-29, miR-126, and let-7 families are abundant in endothelial cells [26], in correlation with our results that showed an increased expression of the let-7 family. MiR-210 plays an important role in angiogenesis. Several studies suggested that upregulated miR-210 could inhibit proliferation and induce apoptosis in endothelial progenitor cells, impairing the angiogenic properties under oxygen and glucose deprivation conditions in endothelial progenitor cells [27]. Wang et al. reported that miR-210 significantly reduced HIF-1 expression, which affects the mobilization of bone marrow endothelial progenitor cells [28]. Moreover, an in vivo study showed that miR-210 decreases the mobilization effect of SDF-1α/CXCL12 on bone marrow-derived angiogenic cells. All of these findings suggest that miR-210 is closely related to the number and function of endothelial progenitor cells [29]. Shi et al. found that miR-375–3p was significantly upregulated following high glucose stress. Functional in vitro assays showed that miR-375 regulates the proliferation, migration, tube formation, and apoptosis of EPCs [30]; in our study the level of miR-375 increased 11-fold compared with AFSCs. The miR-17–92 cluster is a polycistronic miRNA localized in intron 3 of the 13q31.3. chromosome. MiR-17–92 contains six individual miRNAs: miR-17, miR-18a, miR-19a, miR-20a, miR-19b, and miR-92a, which are implicated in apoptosis suppression and angiogenesis induction in tumors. MiR-17–92 is stimulated upon VEGF treatment in endothelial cells. The activation of the miR-17–92 cluster by Elk-1 is necessary for endothelial cell proliferation and vascular sprouting. Laminar flow increased miR-21 and nitric oxide synthesis by engaging the phosphatidylinositol-4,5-bisphosphate 3-kinase/Protein Kinase B Alpha pathway regulating endothelial cell functions. Oscillatory shear flow induces miR-21 expression, stimulating endothelial inflammation mediated by peroxisome proliferators-activated receptor mechanisms [31]. Endothelial cells synthesize endothelin-1 (ET-1), one of the most effective vasoconstrictors, its expression and secretion being stimulated by ischemia, hypoxia, and shear stress. MiR-125a/b is engaged in the regulation of ET-1 factor secreted by endothelial cells. Moreover, its expression has been shown to be regulated by oxidized low density lipoproteins (LDLs) [32]. Also, in our experiments, we observed decreased expression of miR-125b and let-7a, but increased expression of miR-125a in endothelial progenitor cells and avidity for oxidized LDLs uptake. Oxidized LDLs, which are accumulated within the atherosclerotic lesions, enhance the expression of proinflammatory genes, leading to the dysfunction of vascular endothelial cells. Oxidized LDLs cause downregulation of let-7g in endothelial progenitor cells by interaction of the octamer binding transcription factor-1 to the let-7g promoter [33]. Interestingly, both let-7a and let-7b target the lectin-like LDL-receptor1, which is the receptor for ox-LDL in endothelial cells [34].

Smooth muscle differentiation of AFSCs using M231 Medium supplemented with bFGF, EGF, heparin, IGF, and BSA induced significant changes in nine microRNAs from a total of 84 tested; three were downregulated (miR-18a-5p, miR-21-5p, and miR-137,) and six were upregulated (miR-134-5p, miR-155-5p, miR-20a-5p, let-7i-5p, miR-214-3p, and miR-375) compared with undifferentiated AFSCs. Pan et al. showed that miR-137 inhibits vascular smooth muscle cell (VSMC) migration and proliferation. They found that IGFBP-5 (insulin-like growth factor binding protein-5) was a direct target of miR-137 in VSMCs. Also, they demonstrated that MiR-137 overexpression suppressed the activity of the mTOR/STAT3 signaling pathway [35]. Furthermore, miR-18a-5p expression was increased in differentiated VSMCs, whereas it decreased in dedifferentiated VSMCs. Syndecan-4 is a target of miR-18-5p in VSMCs; overexpression of syndecan-4 reduced Smad2 expression, whereas knockdown of syndecan4 increased Smad2 levels in vascular smooth muscle cells [36]. The miR-155-5p functions as an inhibitor in VSMC migration and vascular remodeling in a hypertension state. In addition, miR-155-5p inhibited the levels of BACH1 and suppressed the formation of ROS in SHR VSMCs. Moreover, deletion of miR-155-5p increased the expression of BACH1, and supported the production of superoxide anions, by increased protein expression of NOX2, NOX4, two NAD(P)H oxidases, as the main sources of ROS in cardiovascular cells. Finally, overexpression of miR-155-5p and downregulation of BACH1 repressed the migration of VSMCs. The suppression of BACH1 by miR-155-5p was sufficient to decrease the migration and oxidative stress in VSMCs [37]. A recent study published by Li et al. using a knock-out model of miR-214 in VSMCs investigated the role of miR-214 in vascular smooth muscle cells by quantification of the expression of contractile markers including smooth muscle actin, smooth muscle myosin heavy chain, and calponin. Their results showed that expression of these markers was significantly decreased in miR-214 KO VSMC [38]. Our results are in agreement with these findings, showing that the presence of these contractile markers in differentiated smooth muscle progenitors correlates with an increased expression of miR-214.

Interaction pathway analysis using miRTargetLink2.0 [39] showed that after applying EPC and SPC differentiation protocols the expression of miRNAs was different, with EPCs expressing ten specific miRNAs (hsa-miR-125a-5p, hsa-miR-125b-5p, hsa-miR-181a-5p, hsa-miR-103a-3p, hsa-miR-129-5p, hsa-miR-210-3p, hsa-let-7c-5p, hsa-let-7a-5p, hsa-let-7d-5p, and hsa-let-7e-5p),SPCs only two specific miRNAs (has-miR-20a-5p and has-miR-155-5p), and seven miRNAs (hsa-miR-18a-5p, hsa-miR-13, hsa-miR-134-5, hsa-let-7i-5p, hsa-miR-214-3p, hsa-miR-21-5p, and hsa-miR-375) were common to both EPCs and SPCs (Figure 5). Pathway analysis revealed that hsa-mi-R-210-3p, hsa-mi-R-103-3p, hsa-let-7c-5p, hsa-let-7a-5p, hsa-miR-18a-5p, hsa-miR-125a-5p, and hsa-miR-125b-5p identified in EPCs derived from AFSCs were involved in processes such as cardiovascular development, blood vessel development, and vasculature development, while hsa-mi-R-103-3p was associated only with cardiovascular development and heart morphogenesis (Figure 6). Furthermore, miRNA analysis showed that hsa-miR-125a-5p, hsa-miR-125b-5p, hsa-miR-181a-5p, hsa-let-7c-5p, and hsa-let-7a-5p are involved in both the HIF-1 signaling pathway and response to nitrogen compound signaling (Figure 6), processes that are associated with endothelial cell metabolism.

MiRNA target pathways analysis showed that hsa-miR-20a-5p and hsa-miR-155-5p identified in SPCs were associated with muscle cell proliferation, actin cytoskeleton organization, and organelle formation. The major pathways that orchestrate these processes are the TFG-β and Notch pathways (Figure 7).

## 4. Materials and Methods

### 4.1. Amniotic Fluid Stem Cells Differentiation and Characterization

The amniotic fluid cell cultures were kindly offered by S.C. Personal Genetics S.R.L. Medical Genetic Center Laboratory with the informed consent of the patients and in accordance with the Declaration of Helsinki 1975, amended in 2013, and with national and EU rules. The primary amniotic fluid stem cell cultures were obtained by centrifugation of amniotic fluid at 300× *g* for 10 min. The cells were cultured for 10 days without passages in AmnioMax medium (CN 11269016, Thermo Fischer Scientific, Waltham, MA, USA), with medium changes every 2 days (37 °C, 5% CO_2_, 21% O_2_ in a humidified atmosphere). After 10 days, the primary cultures were passaged and cultured in differentiation medium supplemented with specific growth factors. Differentiation of AFSCs to endothelial progenitor cells (EPCs) was performed by culturing cells in M200 Medium with 10% FBS (fetal bovine serum), 20 ng/mL insulin growth factor (IGF-1), 40 ng/mL vascular endothelial growth factor (VEGF), 10 ng/mL epidermal growth factor (EGF), 10 ng/mL basic fibroblast growth factor (bFGF), 100 μg/mL streptomycin, 100 μg /mL penicillin, and 50 μg/mL neomycin (Thermo Fischer Scientific, Waltham, MA, USA). Smooth muscle cell differentiation of AFSCs was obtained by culture in M231 Medium supplemented with bFGF (2 ng/mL), EGF (0.5 ng/mL), IGF-1 (2 µg/mL), heparin (5 ng/mL), and bovine serum albumin (BSA) (0.2 µg/mL) (Thermo Fischer Scientific, Waltham, MA, USA). The cells were maintained in these media supplemented with endothelial and muscle cell growth factors for 4 weeks, and passaged when reaching subconfluence (37 °C, 5% CO_2_, 21% O_2_ in a humidified atmosphere). The cell culture media was changed 2 times per week [40,41].

### 4.2. Gene Expression and Functional Characterization of Endothelial Progenitor Cells (EPCs)

Expression levels of mRNA were assessed by qRT-PCR. Total RNA was isolated from AFSCs using RNeasy Mini Kit (Qiagen, Hilden, Germany) and reverse-transcription reaction was carried out using M-MLV polymerase (High-Capacity cDNA Reverse Transcription kit, Thermo Fischer Scientific, Waltham, MA, USA). Messenger RNA levels of endothelial related genes (ICAM-1, PECAM-1, eNOS, VE-Cadherin, and vWF) were quantified using TaqMan hydrolysis probes (Thermo Fischer Scientific, Waltham, MA, USA) (Appendix A). Quantitative RT-PCR reactions were performed in a real-time thermocycler (TaqManTM Universal PCR Master Mix, ViiA7, Applied Biosystems, Waltham, MA, USA), respecting manufacturer’s guidelines. The results were expressed using relative quantitation (2^−ΔC_T_), where ΔC_T_ represents the threshold-crossing cycle (CT) difference between values for AFSCs and EPCs. The control for the experiments was represented by the AFSCs cultivated in AmnioMax medium. The immunocytochemistry protocol required the cultivation of EPCs on glass coverslips. For staining, the cells were washed twice with PBS, dried, and fixed with 100% methanol for 20 min. The EPCs were treated with blocking buffer (5% bovine serum albumin in PBS) for 30 min, then incubated overnight, at 4 °C, with primary antibodies: monoclonal IgG mouse anti-human CD31 (1:200; BD Biosciences, La Jolla, CA, USA), polyclonal IgG mouse anti-human CD144 (1:200; Sigma Aldrich, St. Louis, MI, USA), and monoclonal IgG mouse anti-human VEGFR2 (1:100 BD Biosciences, La Jolla, CA, USA). The secondary antibody was added after the cells were washed three times with PBS, for 10 min., and incubated for 2 h, in the dark. The secondary antibody used was goat-anti-mouse Alexa Fluor 488 (1:2000 for CD31 and 1:1000 for CD144 and VEGFR2, Sigma Aldrich, St. Louis, MI, USA) and cell nuclei were stained with Hoechst (1:1000; Sigma Aldrich, St Louis, MI, USA). The slides were washed three times with PBS, and mounted in Vectashield mounting media for fluorescence (Vector Laboratories, Burlingame, CA, USA). Cell preparations were examined using an inverted epifluorescence microscope with an incorporated digital camera system (Axiovert 200 M Carl Zeiss MicroImaging GmbH, Manila, Philippines). *Dil-Acetylated-LDL uptake assay*. EPCs were incubated with 6 μg/mL Dil-Acetylated-LDL-PE (acetylated low-density lipoprotein phycoerythrin conjugated, Thermo Fischer Scientific, Waltham, MA, USA) for 2 h at 37 °C with 5% CO_2_, washed with PBS and fixed with 1% paraformaldehyde (PFA) for 10 min at room temperature. *Ulex europaeus agglutinin (UEA) assay*. EPCs were incubated with 0.01 mg/mL FITC—Ulex europaeus lectin (Sigma-Aldrich, St. Louis, MI, USA) for 2 h, followed by PBS wash. The nuclei of EPCs were counterstained with 1 mg/mL DAPI (Thermo Fischer Scientific, Waltham, MA, USA). The photomicrographs were captured with a digital camera (Digital Net Camera DN100) using an TE300 Eclipse microscope (Nikon, Tokyo, Japan). *Matrigel tube-formation assay*. Evaluation of blood vessel-like networks in Matrigel, was made by seeding the EPCs into 96-well plates at a density of 3000 cells per well. Briefly, 50 μL of Matrigel (Sigma-Aldrich, St. Louis, MI, USA) was added to each well of 96-well plate, and left for gelation 30 min at 37 °C. After polymerization of Matrigel, the EPC suspension was added and incubated for 24 h. Tube formation was assessed using the same microscope-digital camera system.

### 4.3. Gene Expression Assays on Smooth Muscle Progenitor Cells (SPCs)

Messenger RNA levels of muscular-associated genes (smoothelin, calponin 1, α-SMA (α-actin), Myh11 (myosin heavy chain), α-tropomyosin, caldesmon-1, Cav3.1) were quantified using TaqMan hydrolysis probes (Thermo Fischer Scientific, Waltham, MA, USA) (Appendix A).

### 4.4. Patch-Clamp Assays on Smooth Muscle Progenitor Cells (SPCs)

In patch-clamp experiments we used 1.5/0.86 mm diameter borosilicate glass capillaries with inner filament (GC150F-10, Harvard Apparatus, Holliston, MA, USA) pulled in four steps with a PUL-100 microprocessor-controlled equipment (WPI, Sarasota, FL, USA) and fire-polished with a microforge to yield a resistance in solution between 2–3 MΩ. The patch-clamp system included an inverted microscope (IMT-2, Olympus, Tokyo, Japan) placed on an anti-vibratory table in a Faraday cage, a temperature controller (TC202A, Harvard Apparatus, Holliston, MA, USA), a resistive feedback amplifier (WPC-100, ESF electronic, Göttingen, Germany) connected to a Digidata 1322A AD/DA interface controlled by the pClamp8.2 software (Axon Instruments, Molecular Devices, Sunnyvale, CA, USA). The extracellular solution was composed of 1.8 mM CaCl_2_, 135 mM NaCl, 0.9 mM MgCl_2_, 5.4 mM KCl, 10 mM HEPES, 0.33 mM NaH_2_PO_4_, 10 mM D-glucose, pH 7.40 at 25 °C titrated with NaOH. The pipette solution contained 5 mM EGTA, 140 mM KCl, 10 mM HEPES, pH 7.21 at 25 °C titrated with KOH. Some of the experiments were performed in the perforated whole-cell configuration by adding 15 to 30 μM β-escin (Sigma E1378, Kanagawa, Japan) to the pipette solution [42]. SPCs cultured in 24-well plates (some of them pre-incubated with nifedipine 1 μM in the culture medium for 24–48 h) were detached with accutase (Sigma A6964) (7 min at 37 °C), centrifuged, disposed in 35-mm Petri dishes in external solution and kept for 30 min at 37 °C for reattachment. Several voltage protocols were applied: a double voltage ramp protocol (from −120 mV to +80 mV in 2 s and back to −120 mV in the same time, with a holding potential of −70 mV); a standard multi-step voltage protocol to assess several voltage-dependent currents, including T-type and L-type Ca^2+^ currents (*I*_CaT_ and *I*_CaL_) and transient outward K^+^ currents (*I*_to_) (holding potential −80 mV, 300-ms voltage steps from −60 mV to +60 mV in 10 mV increments); a protocol for voltage-dependent Na^+^ currents (*I*_Na_) (holding potential −100 mV, 6-ms voltage steps from −60 mV to +40 mV in 10 mV increments); a voltage protocol to separate *I*_CaL_ from *I*_Na_ based on different voltage dependence of steady-state inactivation [43]. The following pharmacological compounds were applied in individual experiments: the specific BK channel blocker iberiotoxin (STI-400, Alomone Labs, Jerusalem, IL, USA) at 100 nM, the specific *I*_to fast_ (Kv4.2 and 4.3) peptide inhibitor phrixotoxin-1 (Abcam ab141844) at 280 nM, the moderately T-type selective Ca^2+^ channel inhibitor mibefradil (Sigma M5441) at 5 or 10 μM (estimated IC50 for *I*_CaT_ 2.7 μM and for *I*_CaL_ 18.6 μM), the L-type Ca^2+^ channel inhibitor nifedipine (Sigma N7634) at 1 μM, the non-selective anion exchange inhibitor 4, 4′-diisothiocyano-2, 2′-stilbenedisulfonic acid (DIDS) (Calbiochem 309795) at 100 μM, the sulfonylurea receptor inhibitor glibenclamide (Sigma G0639) at 100 μM, the cytochrome inhibitor sodium cyanide (Sigma-Aldrich 380970) at 5 mM, and β-adrenergic receptor agonist isoprenaline (Sigma I5627) at 1 μM.

### 4.5. MiRNA Expression

The expression of miRNA was assessed using The Human Cell Differentiation and Development miScript miRNA PCR Array (Qiagen, Hilden, Germany) that profiles the expression of 84 miRNAs differentially expressed during cellular differentiation and organism development. The data was analyzed using GeneGlobe Data Analysis Center platform (Qiagen, Hilden, Germany).

### 4.6. Data Analysis

Data are described as mean ± SD and mean ± SEM, “*n*” showing the sample number. Statistical significance was evaluated using either Student’s *t* two-tailed test for independent samples or its non-parametric variant (Mann–Whitney test), according to results of normality tests, for quantitative data, and Fisher’s exact probability test for categorical frequency data. For the statistical tests a critical level *p* = 0.05 was set. The datasets produced during this study are available from the corresponding author on reasonable request. MiRNA target pathways analysis was assessed using miRTargetLink2.0. software [39].

## 5. Conclusions

MiRNAs are molecular regulators of cell fate and differentiation during development. Recent advances in molecular biology techniques established miRNA regulatory networks that can provide information on endothelial and muscle cell biology in terms of homeostasis, growth, differentiation, and vascular function. Amniotic fluid stem cells possess specific miRNA expression profiles that modulate stem cell fate. We have identified two miRNAs that were significantly increased in SPCs and ten miRNAs in EPCs and showed their interactions with biological pathways. Our results showed that miRNAs regulate lineage-specific differentiation into endothelial and smooth muscle progenitor cells in AFSCs. These findings can contribute to understanding miRNA regulatory networks facilitating differentiation of stem cells and driving a step forward in regenerative medicine. Further functional studies are necessary to validate these interactions between regulatory miRNA and mRNAs.

## Figures and Tables

**Figure 1 ijms-25-00302-f001:**
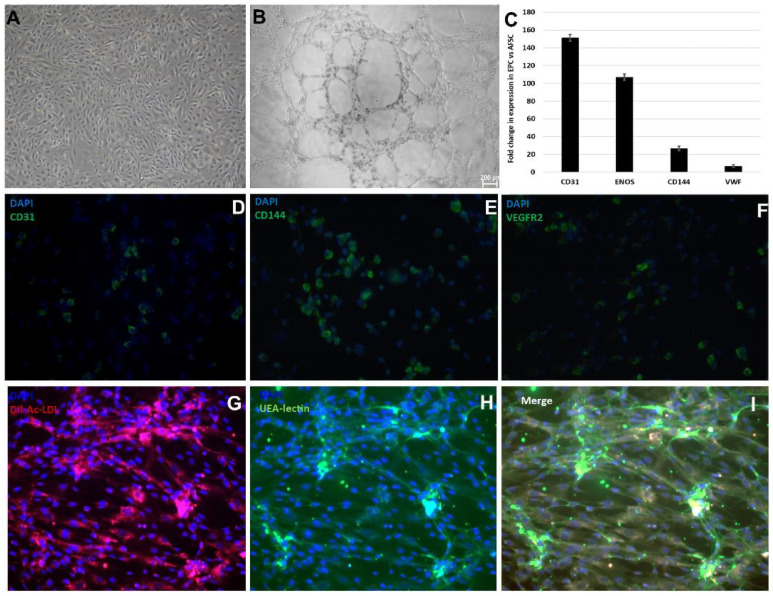
Markers suggestive of endothelial progenitor commitment of AFSCs: (**A**) Morphology of EPCs after 4 weeks of culture with endothelial growth factors; (**B**) Tube-like structures of EPCs on Matrigel extracellular matrix; (**C**) mRNA expression level for human CD31, eNOS, CD144, vWF (TaqMan qRT-PCR, *n* = 4, mean ± SEM, *p* < 0.05, one-way ANOVA); (**D**–**F**) Protein expression level for CD31, CD144, and VEGFR2 (immunohistochemistry using antibodies coupled with Alexa Fluor 488); (**G**) Dil-Acetylated-LDL uptake (red), (**H**) UEA lectin (green), (**I**) merged Dil-Acetylated-LDL and UEA-lectin.

**Figure 2 ijms-25-00302-f002:**
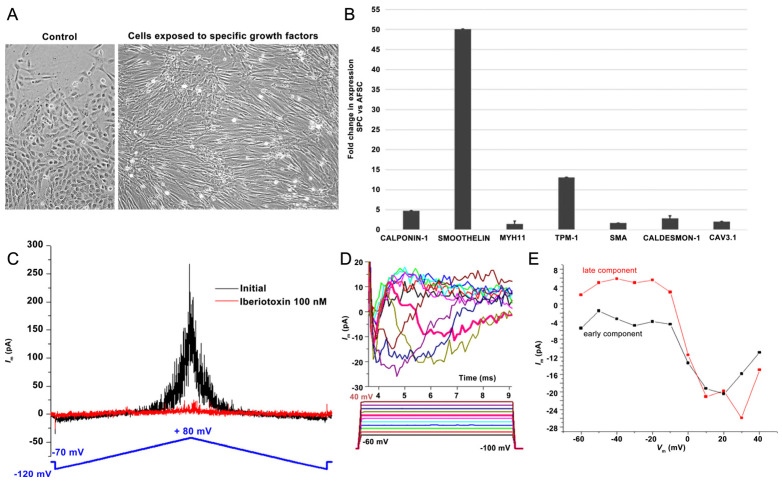
Markers suggestive of smooth muscle commitment of AFSCs. (**A**) Morphology of SPCs after 4 weeks of culture with specific growth factors (bFGF, EGF, heparin, IGF, BSA); (**B**) mRNA expression level for human calponin-1, smoothelin, Mhy-11, TPM-1, SMA, caldesmon-1, Cav3.1 (TaqMan qRT-PCR, *n* = 4, mean ± SEM, *p* < 0.05, one-way ANOVA); (**C**) Double-ramp voltage-clamp protocol recorded before and during application of iberiotoxin 100 nmol/L that blocked big conductance Ca^2+^ -dependent K^+^ (BK) current fluctuations at positive potentials; (**D**,**E**) Inward current components suggestive of *I_CaT_* and *I*_CaL_, and their I–V curves (magnification 20×).

**Figure 3 ijms-25-00302-f003:**
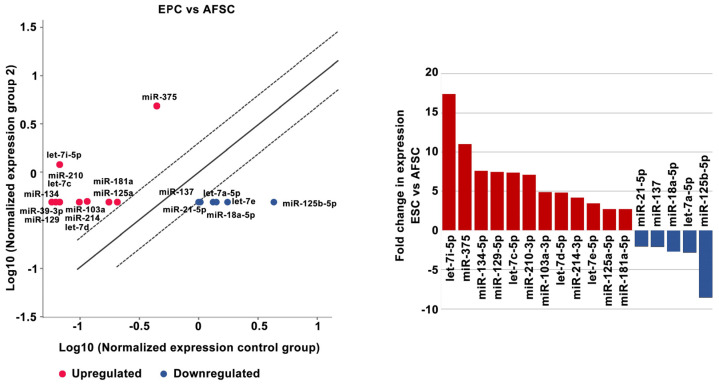
MiRNA expression levels in endothelial progenitor cells after 4 weeks of AFSC differentiation treatment. Fold-change is the normalized miRNA expression in every test sample divided by the normalized miRNA expression in the control sample. Fold-change values greater than 2 are indicated in the diagram as red-upregulated miRNA, blue-downregulated miRNA, *n* = 3, *p* < 0.005.

**Figure 4 ijms-25-00302-f004:**
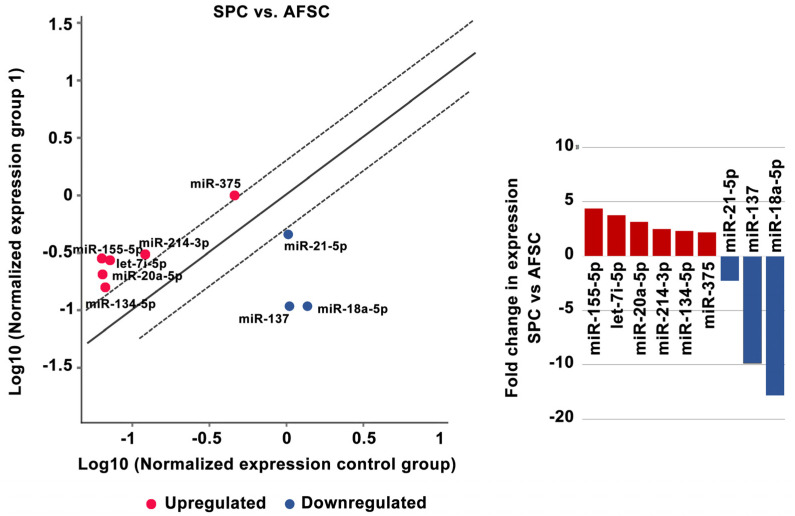
MiRNA expression levels in smooth muscle progenitor cells after 4 weeks of AFSC differentiation treatment. Fold-change is the normalized miRNA expression in each test sample divided by the normalized miRNA expression in the control sample. Fold-change values greater than 2 are indicated in the diagram as red-upregulated miRNA, blue-downregulated miRNA, *n* = 3, *p* < 0.005.

**Figure 5 ijms-25-00302-f005:**
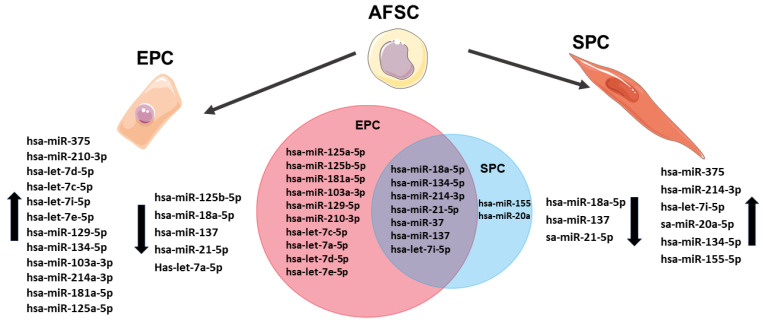
The expression levels of miRNAs in endothelial and smooth muscle progenitor cells.

**Figure 6 ijms-25-00302-f006:**
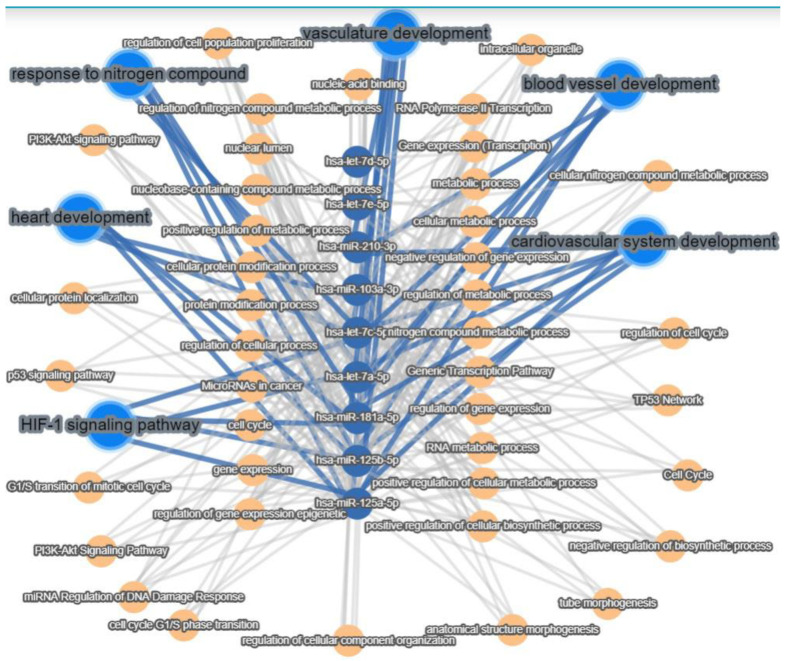
Interaction pathway analysis for hsa-miR-125a-5p, hsa-miR-125b-5p, hsa-miR-181a-5p, hsa-miR-103a-3p, hsa-miR-129-5p, hsa-miR-210-3p, hsa-let-7c-5p, hsa-let-7a-5p, hsa-let-7d-5p, and hsa-let-7e-5p using miRTargetLink2.0 [39]. Blue lines highlight the miRNA target pathways involved in vasculature development and endothelial cell metabolism.

**Figure 7 ijms-25-00302-f007:**
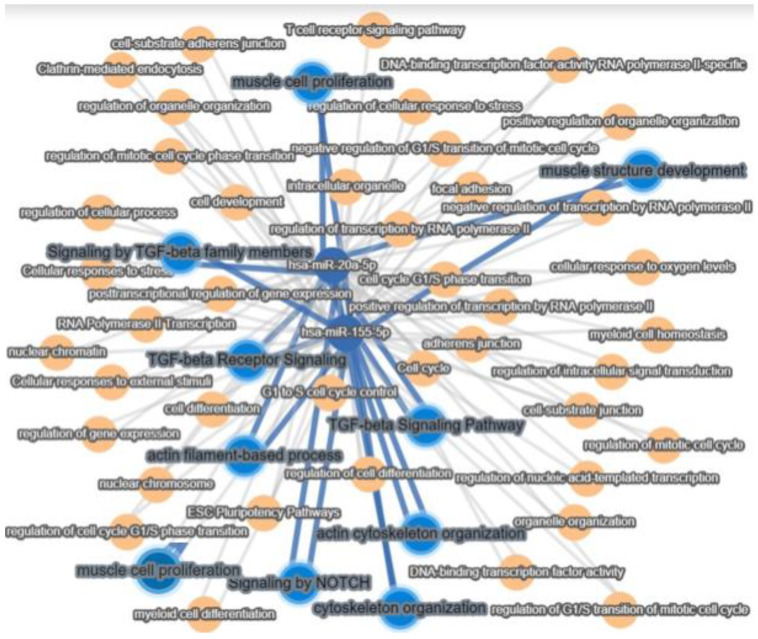
Interaction pathway analysis for hsa-miR-20a-5p and hsa-miR-155-5p using miRTargetLink2.0. Blue lines highlight the miRNA target pathways involved in muscle structure development.

## Data Availability

All data used to support the findings of this study are included within the article or in Appendix A, and they are available upon request from the corresponding author.

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
