# Peer review of "PCR Array Profiling of miRNA Expression Involved in the Differentiation of Amniotic Fluid Stem Cells toward Endothelial and Smooth Muscle Progenitor Cells"

_ijms, 2023, doi:10.3390/ijms25010302_

Round 1
Reviewer 1 Report
Comments and Suggestions for Authors
The study investigates the role of epigenetic modifications in determining the fate of Amniotic Fluid Stem Cells (AFSC) in terms of differentiation into various cell lineages. The focus is on microRNAs (miRNAs) and their unique signatures in contributing to lineage-specific differentiation. The study assesses the miRNA profile in AFSC after 4 weeks of endothelial and muscular differentiation. Results indicate a decreased expression of some miRNAs in Endothelial Progenitor Cells (EPC) compared to undifferentiated AFSC. Additionally, significant changes in other miRNAs are observed during smooth muscle differentiation of AFSC. The study has various shortcomings, both in terms of experimental design and with regard to English language usage. The description of the conducted experiments includes numerous inaccuracies. Often, experiments and data are mentioned without corresponding explanations in the text, especially regarding the treatment of AFSCs. One key observation derived from reviewing the manuscript is the lack of clarity regarding the treatments administered to AFSCs to prompt their differentiation into the endothelial and smooth muscle phenotypes. This holds significant importance in comprehending the study's rationale. Despite the author's insistence on labeling cells supposedly derived from AFSCs as EPC and smooth muscle cells, the true nature of these cells, in my opinion, is not emerging from the study. I have listed below my main concerns about the study.
Major.
· Figure 1. The results presented in Figure 1 are confounding. The authors assert that AFSC, following a 24-hour treatment period with unspecified endothelial growth factors, exhibit a morphology reminiscent of endothelial cells. In my view, these conclusions lack support from the experimental data depicted in Figure 1. Specifically, in panel A, the (spindle-like) cell morphology more closely resembles that of mesenchymal cells or fibroblasts rather than endothelial cells. Additionally, the significance of the results in panel C remains unclear. What serves as the reference (control) in this instance? This must be specified in the text. The authors need to establish convincingly that these cells have indeed adopted an endothelial phenotype. Therefore, if the objective was to illustrate the differentiation of AFSCs into an endothelial phenotype, I recommend employing multiple approaches. For example, in addition to PCR analysis, use immunofluorescence staining to detect the expression of endothelial cell-specific markers, CD31, vWF, and VE-cadherin. The same can be done by analyzing these cell surface markers expression using flow cytometry. If authors intend to illustrate the endothelial phenotype, an elegant method would be also to measure von Willebrand Factor secretion through ELISA. Regarding panels D and the following ones (that are not labeled), please be aware that the absorption of acetylated low-density lipoprotein is not exclusive to vascular endothelial cells; macrophages, utilize this mechanism to engulf cholesterol, too.
· Figure 1, line 93. What does the statement "…these factors control EPC proliferation, adherence, differentiation, and survival (Figure 1C)" have to do with figure 1C?
· Line 101. The statement "Stimulation of AFSC with FGF, EGF, heparin, IGF, and BSA promoted differentiation toward smooth muscle cells, inducing some phenotypic changes such as spindle shape morphology and elongated actin-rich protrusions could be observed in Figure 1C" has no experimental evidence. Where do the authors demonstrate these data? Where do authors run experiments with the FGF, EGF, heparin, IGF? This is not clear.
· Figure 2. Please be aware that the panels in Figure 2 are incorrectly labeled. Furthermore, it is unclear which "growth factors" were employed in treating AFSC before investigating their differentiation into muscle cells. No data regarding these experiments are available.
· Figure 2. What about the right bottom panel of Figure 2? It is not described.
· Figure 3. This figure simply shows the results of an investigation into the expression of microRNAs (miRNAs) during Endothelial and Muscular Cell Differentiation. The authors report that, after a 4-week, several miRNAs, including miR-125a-5p, miR-181a-5p, and miR-134-5p, exhibited increased expression, while others, such as let-7e-5p and miR-125b-5p, displayed decreased expression. However, the authors confine themselves to listing the data obtained without delving into the significance of these findings. It is crucial for the authors to elucidate the biological importance of these miRNAs in the differentiation of AFSCs, at least for some of them. The same applies to figure 4.
· Discussion. Regarding the discussion, the primary emphasis should be on examining the obtained results. In this instance, it appears more as a simple recapitulation of results already documented in the literature. There is minimal exploration of potential speculations and hypotheses from the authors regarding the significance and biological implications that these findings might entail.
Minor.
· I believe the title contains some typos. I would suggest rephrasing and correct it. It may be perhaps “PCR Array Profiling of miRNA Expression Involved in the Differentiation of Amniotic Fluid Stem Cells Toward Endothelial and Smooth Muscle Progenitor Cells"
· Line 47. The statement: “When grown in hydro gels, AFSC have demonstrated the ability to form capillary-like networks” requires citing an appropriate reference.
· Line 86. Please elucidate the precise meaning of the terms "endothelial-differentiation growth factors" and "endothelial-specific medium" as intended by the author.
· What does “SPC” through the text stand for?
· Many abbreviations or acronyms in the text are not explicitly defined, and some readers may not be familiar with them. This makes the reading of the text challenging (e.g., SPC, VSMC, etc.).
· Line 13-14. Please rephrase the sentence “There is a tight correlation among DNA methylation, histone modifications, and small noncoding RNAs during the epigenetic control of stem cells differentiation”, copied from the following source https://pubmed.ncbi.nlm.nih.gov/30627172

Comments on the Quality of English LanguageThe study has several deficiencies in terms of English language usage. It must be extensively revised
Reviewer 2 Report
Comments and Suggestions for Authors
In the current manuscript, the authors have investigated the role of miRNAs in the differentiation of amniotic fluid stem cells (AFSC) and address how their unique signatures contribute to lineage-specific differentiation. Authors show AFSC differentiation into endothelial and smooth muscle progenitor cells and report the expression of miRNAs involved in differentiation.
The article is well structured into sections and subsections. The introduction is comprehensive and well written. The article is within the scope of the journal.
However, there are some concerns that need to be addressed to improve the article:
1) Page 3, line 103: Based on figure 1, the morphology cells is presented in the panel 1A, while panel 1C exhibits the mRNA expression levels. Therefore, the following reference needs correction. “…phenotypic changes such as spindle shape morphology and elongated actin-rich protrusions could be observed in Figure 1C.”
2) Page 4, figure 2, line 129-132: The description in the legend for panels (A and B) does not correspond to the image in the respective panels. Panel A shows the morphology of cells and Panel B depicts the mRNA expression level not vice versa.
3) Page 10, line 377-388: The sentences from instruction to authors needs to be removed.
4) Page 10, line 390: The data availability statement needs rephrasing for clarity.
5) Supplementary Figures: There is a typing error that needs correction, “Figura” needs to be corrected to figure.
Comments on the Quality of English LanguageOverall the quality of English Language is fine. There are some sentences that need rephrasing for clarity.
Reviewer 3 Report
Comments and Suggestions for Authors
The authors assessed miRNAs profile using miScript miRNA PCR Array in amniotic fluid-derived mesenchymal stem cells (AFSC) after 4 weeks of endothelial and muscular differentiation. Major revision is suggested based on the following issues:
1. The expressions of surface antigens should be detected by flow cytometry to verify the successful isolation of amniotic stem cells. For example, positive for CD90, CD44 and negative for CD45 and CD31.
2. Besides mRNA expression, western blot is suggested to confirm the successful differentiation of AFSC in endothelial and smooth muscle progenitor cells.
3. Also, gene ontology and pathway analysis can be performed for an in-depth study.
4. Those citations involving miRNA in differentiation of AFSC [ref 11-16] were published in 2012 to 2016. More recently published works should be cited and compared with the proposed manuscript. For example, two recent works (J Cell Biochem. 2020, 121(2):1811-1822; Cell Biol Int. 2019, 43(3):299-312) have already profile the epigenetic alterations including the expression of lineage-specific genes, microRNAs and chromatin modifying proteins by RT-qPCR and Western blot, respectively.
5. In Line 357-367 (the Conclusions Part), amniotic fluid was not mentioned. And the authors did not summarize their findings or observations in Conclusion Part. Based on these findings, what possible future studies can be performed?
6. Please double check the manuscript. For example, in Line 103, “Figure 1C” should be “Figure 2A”. The format of Refences should be unified. Many abbreviations lacked their full names when they first appeared, such as “AFSC” in Abstract, EPC, SPC and so on.
Comments on the Quality of English Language
Can be improved.
Round 2
Reviewer 1 Report
Comments and Suggestions for Authors
Reply to authors (ms. ijms-2735368)
Authors addressed many of the comments raised during the revision process. However, I would like to point out the following issues that, in my opinion, represent a weakness of the study. Regarding my previous comment inquiring about proving that AFSC has adopted an endothelial phenotype, the authors responded, stating, "We performed flow-cytometry on these type of cells in our previous work. These are the published papers in which we characterized the 'AFSC' and 'EPC.” While this is reasonable, it is important to note that this study is independent and there is no guarantee that a different preparation (isolation) of 'AFSC' will exhibit the same behavior. Furthermore, conducting the suggested analysis would not only corroborate but also enhance the support for the previous findings by the same authors. Alternatively, in addition to referencing their previous studies, it would be fair for the authors to assert that the differentiation of ECs from AFSC is a highly reproducible and statistically significant phenomenon persisting across various AFSC preparations. I believe that with these few adjustments, the manuscript is suitable for publication in IJMS.
· Figure 1C. The IF inset shown in the panel C of Figure 1 is misplaced. I would suggest to make a separate panel.
· Figure 2A. Figure legend reads: “Morphology of SPC after 4 weeks of culture with specific growth factors (bFGF, EGF, heparin, IGF, BSA). The authors state in the text that “Stimulation of AFSC with EGF (epidermal growth factor), bFGF (basic fibroblast growth factor), heparin, IGF (insulin growth factor), and BSA (bovine serum albumin) promoted differentiation toward smooth muscle cells, inducing some phenotypic changes such as spindle shape morphology and elongated actin-rich protrusions could be observed in Figure 2A". Please note that a panel showing untreated (time 0) cells is required. How can readers judge the outcome of treatments without a control image?
· Figure 7. Image quality need to be improved for better readability.

Comments on the Quality of English LanguageExtensive editing of English language required
Author Response
We appreciate the time and effort that you have dedicated to providing your valuable feedback on the paper. The suggestion will really improve the quality of the manuscript.
Authors addressed many of the comments raised during the revision process. However, I would like to point out the following issues that, in my opinion, represent a weakness of the study. Regarding my previous comment inquiring about proving that AFSC has adopted an endothelial phenotype, the authors responded, stating, "We performed flow-cytometry on these type of cells in our previous work. These are the published papers in which we characterized the 'AFSC' and 'EPC.” While this is reasonable, it is important to note that this study is independent and there is no guarantee that a different preparation (isolation) of 'AFSC' will exhibit the same behavior. Furthermore, conducting the suggested analysis would not only corroborate but also enhance the support for the previous findings by the same authors. Alternatively, in addition to referencing their previous studies, it would be fair for the authors to assert that the differentiation of ECs from AFSC is a highly reproducible and statistically significant phenomenon persisting across various AFSC preparations. I believe that with these few adjustments, the manuscript is suitable for publication in IJMS.
We use in these experiments the same cells that we used also in experiments for differentiation. After differentiation we tested also for the presence of miRNAs, that was the reason that we said that are characterized in previous work.
- Figure 1C. The IF inset shown in panel C of Figure 1 is misplaced. I would suggest making a separate panel.
Thank you for this observation, we put the images in a separate panel.
- Figure 2A. Figure legend reads: “Morphology of SPC after 4 weeks of culture with specific growth factors (bFGF, EGF, heparin, IGF, BSA). The authors state in the text that “Stimulation of AFSC with EGF (epidermal growth factor), bFGF (basic fibroblast growth factor), heparin, IGF (insulin growth factor), and BSA (bovine serum albumin) promoted differentiation toward smooth muscle cells, inducing some phenotypic changes such as spindle shape morphology and elongated actin-rich protrusions could be observed in Figure 2A". Please note that a panel showing untreated (time 0) cells is required. How can readers judge the outcome of treatments without a control image?
We add an image with cells before the treatment (time 0) with growth factors.
- Figure 7. Image quality needs to be improved for better readability.
We improve the quality of the image for images.
Reviewer 3 Report
Comments and Suggestions for Authors
The authors have addressed most of the comments submitted. Therefore, I consider that this manuscript deserves publication in this Journal after addressing the following minor concerns:
1. Question 1 first revision: the authors added the flow cytometry data in the Supplementary Materials which have already been published in their previous work (Arch Balk Med Union. 2022;57(1):8-16). To avoid self-plagiarism, these data should be deleted or used under permission and citation.
2. The current length in Conclusion Part (Line 557-561) was too short. Please augment the contents.
Comments on the Quality of English LanguageNo comments.
Author Response
We appreciate the time and effort that you have dedicated to providing your valuable feedback on the paper. The suggestion will really improve the quality of the manuscript.
- Question 1 first revision: the authors added the flow cytometry data in the Supplementary Materials which have already been published in their previous work (Arch Balk Med Union. 2022;57(1):8-16). To avoid self-plagiarism, these data should be deleted or used under permission and citation.
We deleted the flow-cytometry figure in the Supplementary Materials.
- The current length in Conclusion Part (Line 557-561) was too short. Please augment the contents.
We add more comments in the Conclusion part.
Round 3
Reviewer 1 Report
Comments and Suggestions for Authors
The authors have responded to previous comments. The manuscript quality has improved, and it is suitable for publication.